# Combining machine learning and nanopore construction creates an artificial intelligence nanopore for coronavirus detection

Masateru Taniguchi [1 ✉], Shohei Minami[2], Chikako Ono[2,3], Rina Hamajima[2], Ayumi Morimura[4], Shigeto Hamaguchi[4,5], Yukihiro Akeda[2,4,5], Yuta Kanai[2], Takeshi Kobayashi[2], Wataru Kamitani[6], Yutaka Terada[7], Koichiro Suzuki[8], Nobuaki Hatori[8], Yoshiaki Yamagishi [4,5,9], Nobuei Washizu[10], Hiroyasu Takei[11], Osamu Sakamoto[11], Norihiko Naono[11], Kenji Tatematsu[1], Takashi Washio[1], Yoshiharu Matsuura [2,3 ✉] & Kazunori Tomono[4,5 ✉]

High-throughput, high-accuracy detection of emerging viruses allows for the control of disease outbreaks. Currently, reverse transcription-polymerase chain reaction (RT-PCR) is currently the most-widely used technology to diagnose the presence of SARS-CoV-2. However, RT-PCR requires the extraction of viral RNA from clinical specimens to obtain high sensitivity. Here, we report a method for detecting novel coronaviruses with high sensitivity by using nanopores together with artificial intelligence, a relatively simple procedure that does not require RNA extraction. Our final platform, which we call the artificially intelligent nanopore, consists of machine learning software on a server, a portable high-speed and high-precision current measuring instrument, and scalable, cost-effective semiconducting nanopore modules. We show that artificially intelligent nanopores are successful in accurately identifying four types of coronaviruses similar in size, HCoV-229E, SARS-CoV, MERS-CoV, and SARS-CoV-2. Detection of SARS-CoV-2 in saliva specimen is achieved with a sensitivity of 90% and specificity of 96% with a 5-minute measurement.

[1] The Institute of Scientific and Industrial Research, Osaka University, Ibaraki, Osaka, Japan. [2] Research Institute for Microbial Diseases, Osaka University, Suita, Osaka, Japan. [3] Center for Infectious Diseases Education and Research, Osaka University, Suita, Osaka, Japan. [4] Graduate School of Medicine, Osaka University, Suita, Osaka, Japan. [5] Osaka University Hospital, Osaka University, Suita, Osaka, Japan. [6] Graduate School of Medicine, Gunma University, Maebashi, Gunma, Japan. [7] Center for Vaccine Research, University of Pittsburgh, Pittsburgh, PA, USA. [8] The Research Foundation for Microbial Diseases of Osaka University, Suita, Osaka, Japan. [9] Medical Center for Translational and Clinical Research, Osaka University Hospital, Osaka University, Suita, Osaka, Japan. [10] ADVANTEST Corporation, Kazo, Saitama, Japan. [11] Aipore Inc., Shibuya, Tokyo, Japan. ✉email: taniguti@sanken.osaka-u.ac.jp; matsuura@biken.osaka-u.ac.jp; tomono@hp-infect.med.osaka-u.ac.jp

Human coronavirus, HCoV-229E, is one of the first coronavirus strains reported to be associated with nasal colds[1]. In the past 20 years, other species of coronaviruses, namely, severe acute respiratory syndrome (SARS) and Middle East respiratory syndrome (MERS), have caused a pandemic in the form of severe respiratory illness[2]. Recently, SARS-CoV-2, the seventh species of coronavirus, has spread all over the world, causing the outbreak of an acute respiratory disease[3–10]. As well as vaccines and effective treatments, testing, and quarantine are needed to control the transmission of the virus. Currently, RT-PCR is the gold standard for SARS-CoV-2 testing that is based on the principle that the single-stranded RNA is present in this virus and the primer forms a double helix. Prior to the genetic test, the process of extraction and purification of viral RNA is time-consuming. The exposure increases the risk of the inspector contracting the virus. Therefore, there is a need for an inspection method with higher throughput[11].

Nanopores have through holes with diameter ranging from several nanometers to several hundreds of nanometers on the substrate[12–18]. Low-aspect solid-state nanopores with nanopore thickness/diameter <1 are used for the detection of DNA[19], viruses[19–22], and bacteria[23]. When the virus is transported from the *cis* side to the *trans* side by electrophoretic force, the ionic current decreases (Fig. 1a, b). The ionic current versus time waveform obtained from the nanopore has information on the volume, structure, and surface charge of the target being analyzed[23]. At the laboratory level, it has been demonstrated that by classifying the waveform data using artificial intelligence, a single virus can be directly identified with high accuracy that does not require the extraction of the genome[20,23–28]. However, in order to obtain sufficient amount of virus learning data in clinical specimens and achieve highly precise detection with increased reproducibility, the manufacturing of nanopore devices with improved accuracy and better yield is a critical constraint. On the other hand, the ionic current obtained from the nanopore is in the order of several tens of pA. The current characteristics obtained from the nanopore largely depend not only on the electrical characteristics of the nanopore, but also on the electrical characteristics of the measuring device and the fluidic device that transports the specimen to the nanopore[17]. Therefore, the development of a dedicated measuring device and flow channel suitable for the nanopores has also been a major limitation in realizing a highly accurate diagnostic system. In the current study, we have developed an artificial intelligence-assisted nanopore-based device to accurately detect the viruses.

## Results

**AI-Nanopore platform**. We developed a nanopore module (25 mm × 25 mm × 5 mm) in which a nanopore chip and a plastic channel were fused (Fig. 1c). Both sides of the silicon chip were chemically bonded to the plastic channel. The hydrophilic channels on the front surface (blue) and the back surface (red) revealed a crossbar structure and passed through the nanopores. From the specimen inlet with a diameter of 1 mm, 15 μl of buffer solution and specimen were pipetted into the front and back channels, respectively. Ag/AgCl electrodes were fabricated on the polymer substrate in each channel for stable current measurement with high reproducibility for a longer duration. A silicon chip (5 mm × 5 mm × 0.5 mm) on which a 50-nm thick SiN was deposited had nanopores about 300 nm in diameter comparable with the diameters of coronaviruses of about 80–120 nm (Fig. 1d, e)[1,29]. Silicon chips were manufactured in units of 12-inch wafers by using microfabrication technology, and were cut into chips by dicing. Through this mass production process, nanopores were produced with high accuracy (diameter error ±10 nm) and high yield (90%).

The diameter of the nanopore was flexible and was modified according to the size of the virus to be measured.

The performance of the developed measurement system is evaluated using standard polystyrene nanoparticles nearly uniform shape and average diameters of 200 and 220 nm that are diluted with 1× phosphate buffered saline (PBS). A buffer solution is placed in the *trans* channel of the nanopore module, nanoparticles of 15 μl diluted with the buffer solution is placed in the *cis* channel, and the module is placed in a dedicated cartridge. The cartridge is placed in a measuring instrument and the current–time waveform is measured for 0.1 V applied voltage. Nanoparticles have been measured using portable nanoSCOUTER[TM]. The number of nanopore devices used and the corresponding waveforms obtained are shown in Supplementary Tables S1–S3, respectively.

Waveforms of the two nanoparticles are obtained with high reproducibility (Fig. 1f–h). The histograms of $I_p$ and $t_d$ obtained from the waveforms corresponding to the nanoparticles have regions that overlap with the histogram (Fig. 1i, j). This overlap hinders the high-precision identification of the two nanoparticles using a single waveform. In order to overcome this issue, machine learning is implemented based on the algorithm shown in Fig. 1k. The detailed machine learning algorithm is described in Supplementary Figs. S1–S5. In the data training phase of the machine learning algorithm, the ionic current–time waveform of each nanoparticle is automatically extracted by an in-house developed signal extraction software and the features of the two nanoparticles are created. In the case of nanoparticles, the features are independent parameters such as $I_p$, $t_d$, current vector, time vector, and a combination of these features (e.g., $I_p + t_d$). The current vector ($I_1, I_2, \cdots, I_{10}$) and time vector ($t_1, t_2, \cdots, t_{10}$) are obtained by dividing the waveform data into tenfolds along the time and current directions[30]. All the features are merged and machine learning is performed to create a single classifier with the highest $F$-value (Supplementary Fig. S5). Precision in machine learning refers to the percentage of data that is expected to be positive and is actually positive, whereas recall refers to the percentage of data that is actually positive and is predicted to be positive. Since precision and recall are in a tradeoff relationship, an index that considers these two indexes together, i.e., the harmonic mean of the two indexes, is defined as the $F$-value. The $F$-value is defined by Equation (4) in Supplementary Fig. S5 and is calculated using the confusion matrix. The discrimination accuracy between the 200 and 220 nm nanoparticles using an ionic current–time waveform is 97% (Fig. 1l). This indicates that a single ionic current–time waveform is sufficient to precisely differentiate the nanoparticles. The classifier has been built using the measurement data obtained from the three modules that is unaffected by the manufacturing variations of the nanopores and the measurement environment (Supplementary Figs S6 and S7). The accuracy of 97% for a single waveform indicates that if two waveforms are acquired, a classification accuracy of ≥99.9% can be achieved.

**Discriminating cultivated coronaviruses**. When HCoV-229E viral sample of 100 pfu/μl is measured at −0.1 V, the ionic current–time waveforms are obtained at a frequency of 14.2 pulses/min (Fig. 2a, b). When measured at 0.1 V, one waveform is obtained in 15 min. To confirm that the virus passes through the nanopore at −0.1 V, RT-PCR measurement is performed on the solution in the *trans* channel. When the solution is extracted from the *trans* channel, it is expected that the virus will be adsorbed on the wall surface of the channel and the number of viruses that are extracted reduces. Hence, RT-PCR measurement is performed after acquiring about 1200 waveforms. The test results showed the

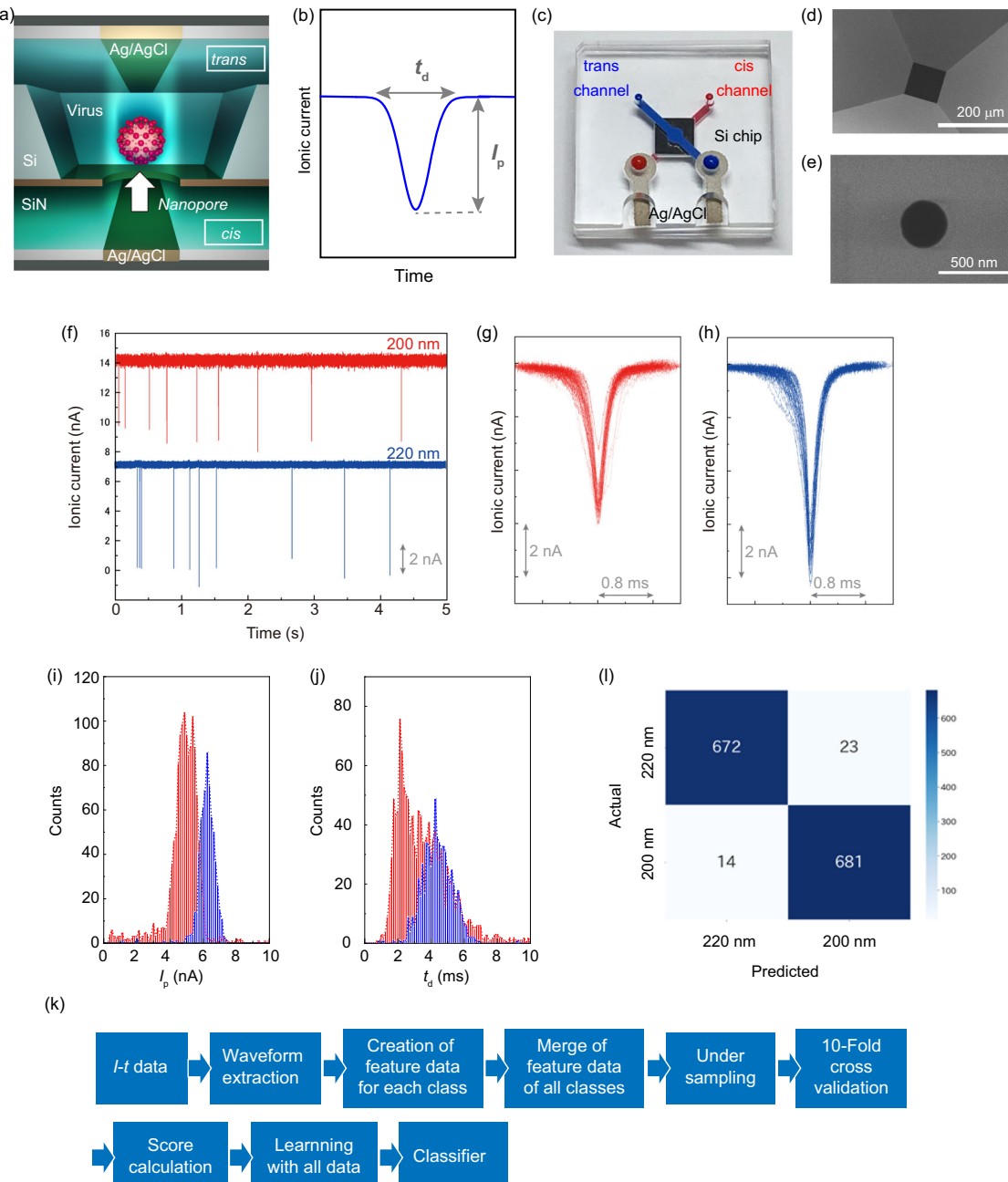

**Fig. 1 Basic characteristics of AI-nanopore platform. a** Nanopore structure fabricated in silicon nitride (SiN) membrane on a silicon (Si) substrate. A specimen and a buffer are placed in the *cis* and *trans* channels. Silver–silver chloride (Ag/AgCl) electrodes are placed on both sides of the substrate. **b** Ionic current flows when voltage is applied. $I_p$ and $t_d$ show the peak current and duration of the current of an ionic current waveform, respectively. **c** An optical photographic image of the nanopore module. Red and blue are *cis* and *trans* channels, respectively. The black central part is a silicon chip with nanopores. **d**, **e** Scanning electron microscope image of the nanopore entrance and the nanopore with a diameter of 300 nm seen from the *trans* side. The black region in the center of the image is a hollow SiN thin film. Six times experiments were repeated independently for the observations. **f** Ionic current–time traces of nanoparticles with 200 and 220 nm diameter. **g**, **h** Ionic current–time profiles, which show an overlap of 100 waveforms each, obtained by nanoparticle measurements. The color code corresponds to **f**. **i**, and **j**, Histograms of $I_p$ and $t_d$ of nanoparticles. The color code corresponds to **f**. **k**, Machine learning algorithm used to identify the nanoparticles and cultured viruses. **l** Confusion matrix of nanoparticles obtained by machine learning. The numbers in the matrix represent the number of waveforms obtained by the measurements. The color bar indicates that the darker the blue, the greater the number of pulses. The number of the color bar indicated the number of waveforms. Source data of **f**–**j** and **l** are provided as a Source Data file.

viral presence when passed through the nanopore at −0.1 V. However, in the RT-PCR performed in the *trans* channel that is left standing for 6 h without applying a voltage, absence of virus is noted. This result has demonstrated that the efficient viral transmission through the nanopore is achieved by electrophoresis instead of diffusional movement.

To investigate the detection limit of the nanopore platform, the ionic current–time waveform is measured at −0.1 V by varying the HCoV-229E concentration (Fig. 2c). The threshold of the viral concentration is set at 250 pfu/μl due to the difficulty in culturing it at high concentrations. When the number of waveforms obtained by measuring for 15 min is examined, an average of three

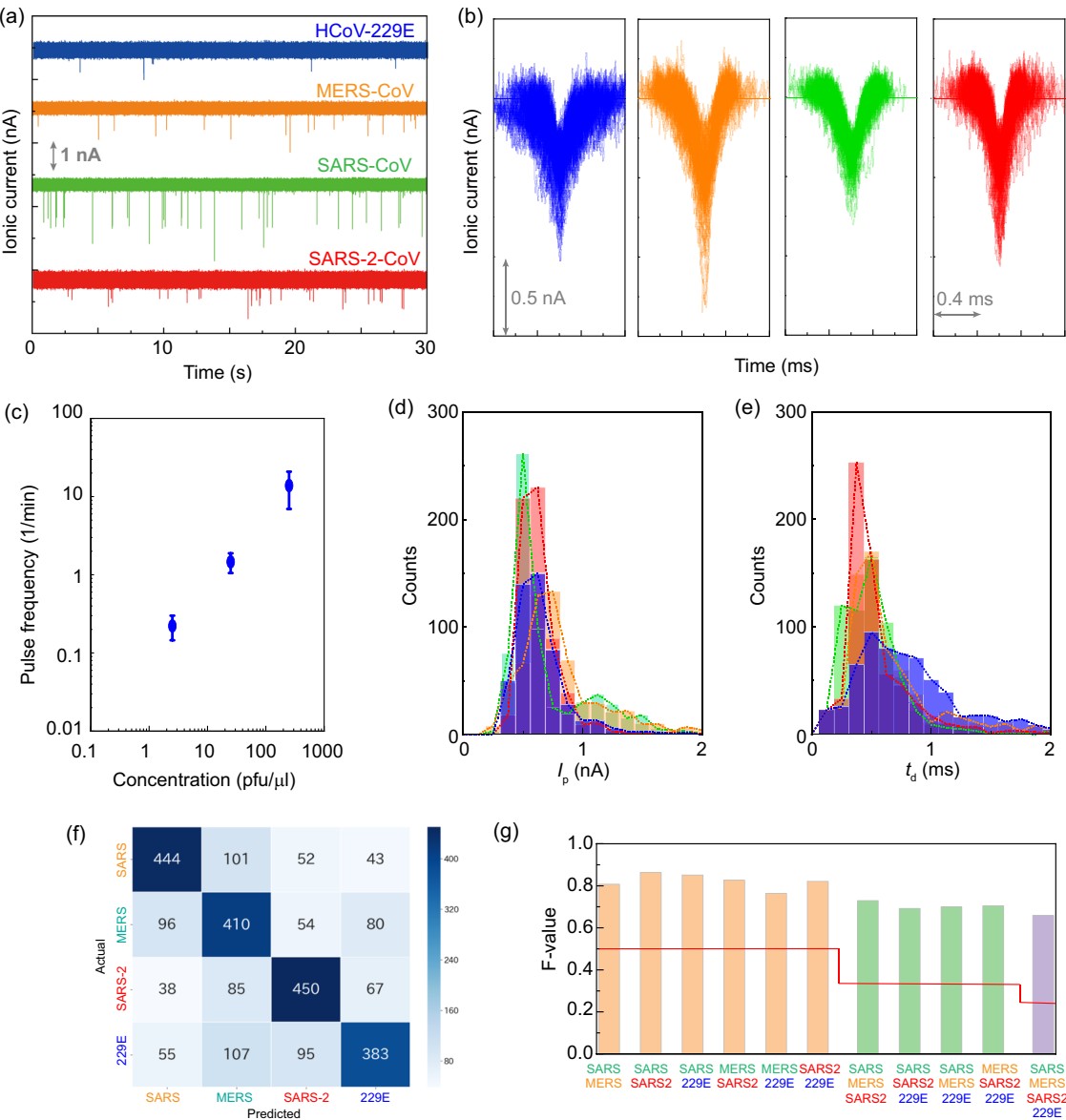

**Fig. 2 Identification of cultured coronavirus. a** Ionic current–time traces of cultured HCoV-229E, MERS-CoV, SARS-CoV, and SARS-CoV-2. **b** Ionic current–time profiles, which are obtained by measuring the cultured coronaviruses and overlay 100 waveforms. The color code corresponds to **a**. **c** Dependence on the number of detected waveforms on the concentration of HCoV-229E. The number of waveforms per unit minute was calculated based on the number of waveforms obtained during the measurement time of 15 min. Three independent samples examined over three independent experiments. Data are presented as mean values ± SD. **d**, **e** Histograms of $I_p$ and $t_d$ of cultured coronaviruses. The color code corresponds to **a**. **f** Confusion matrix obtained by machine learning corresponding to the four cultured coronaviruses. The numbers in the matrix indicate the number of waveforms. The machine learning algorithm follows Fig. 1k. The color bar indicates that the darker the blue, the greater the number of pulses. The number of the color bar indicated the number of waveforms. **g** Identification accuracy in combinations of two, three, and four types of coronaviruses. Each virus is identified when the identification accuracy of the red line or higher is obtained. Orange, green, and purple correspond to two, three, and four virus-identifying F-values, respectively. SARS, SARS2, MERS, and 229E represent SARS-CoV, SARS-CoV-2, MERS-CoV, and HCoV-229E, respectively. Source data of **a**, **b**, and **d**–**g** are provided as a Source Data file.

waveforms could be obtained at 2.5 pfu/μl. However, for the same duration with a concentration of 0.25 pfu/μl, no waveform is obtained. It is therefore concluded that the detection limit of coronavirus in Dulbecco's modified Eagle's medium (DMEM) is 2.5 pfu/μl when 0.1 V is applied for 15 min using a nanopore with a diameter of 300 nm.

The ionic current–time waveforms of MERS-CoV, SARS-CoV, and SARS-CoV-2 are individually measured under the same experimental conditions as HCoV-229E (Fig. 2a, b). The number of nanopore modules used in the measurement and the number

of waveforms obtained are shown in Supplementary Table S4. The histograms of $I_p$ and $t_d$ of the four viruses reveal a major overlap (Fig. 2d, e), which complicates the accurate identification of the viruses. Machine learning is performed to identify the four cultured viruses using the algorithms (Fig. 1k and Supplementary Figs S1–S5) and features that are used to identify the nanoparticles. A classification model is created by using random forest algorithm on the waveforms of the four coronaviruses. The F-value for the four types of viruses is 0.66 (Fig. 2f, g). When this value is greater than 0.25, the four viruses could be identified in

one waveform. The results show that the artificial intelligent nanopores can identify coronaviruses with high accuracy. The identification accuracy of the combination of two viruses and that of three viruses showed higher $F$-values $\geq 0.76$ and $\geq 0.69$, respectively (Supplementary Figs S8 and S9). The discrimination results have also revealed that the MERS-CoV and HCoV-229E species are the most difficult to discriminate.

**Diagnostics for clinical specimens.** Due to the high viscosity of saliva, it was filtered through a 0.45-μm membrane filter and diluted with 1× PBS. When the ionic current–time waveforms of the PCR-positive specimens of saliva are measured for SARS-CoV-2, the waveform is obtained at both 0.1 and −0.1 V with ten times higher number of waveforms obtained at the former than the latter applied voltage (Fig. 3a). To confirm that the novel coronavirus passes through the nanopore at 0.1 V, RT-PCR is performed on the solution in the *trans* channel after obtaining about 3000 waveforms. RT-PCR measurements have demonstrated that the new coronavirus passed through the nanopore when 0.1 V is applied. This result indicates that the surface charges of the cultured virus and its counterpart in the clinical specimen are different[31].

Ionic current–time traces of PCR-positive and PCR-negative specimens of saliva both showed pulsed waveforms (Fig. 3a). The number of nanopore modules used and the corresponding waveforms obtained are shown in Supplementary Table S5. The histograms of $I_p$ and $t_d$ obtained by measuring saliva completely overlap (Fig. 3b, c). Therefore, it is not possible to accurately determine whether a specimen is positive or negative based on the histograms.

Alternatively, using machine learning, waveforms were extracted from the ionic current–time traces of PCR-positive and PCR-negative specimens to determine positive and negative of the new coronavirus (Fig. 3d, e and Supplementary Figs S10–S12). Features similar to the studies on nanoparticles and cultured viruses are generated for each sample. The waveform obtained from the PCR-negative specimen is a noise waveform. Assuming that the noise waveform is common to the PCR-negative specimen and the PCR-positive specimen, the waveform obtained by the PCR-positive specimen is composed of the noise waveform and the waveform of the new coronavirus. When the assembly including the positive unlabeled classification method[30,32] is used, the waveform group of the new coronavirus and the waveform group of the noise from the waveform group of the PCR-negative specimen are learned from the waveform group of the PCR-positive specimen. The positive ratio given by the ratio of the number of waveforms of the new coronavirus to the number of waveforms of the specimen is calculated. The positive ratio provides a threshold for determining whether a specimen is positive or negative. Comparing the waveform corresponding to the novel coronavirus and that pertaining to noise, it is learned whether a given waveform is a positive or negative waveform. This learning empowers the classifier that results in the highest discrimination accuracy and confidence corresponding to one waveform. Confidence is a measure of the level of accuracy with which a waveform is classified as positive or negative. In clinical specimens, random forest algorithm based classifier gives the highest accuracy. In the diagnosis of clinical specimens, the algorithm shown in Fig. 3e and Supplementary Fig. S12 is used to determine the positive or negative nature of a specimen. The confidence is calculated from the $F$-value computed by using the classifier obtained by machine learning. Subsequently, the positive and negative nature of a waveform is determined. The positive and negative of the specimen are determined based on the threshold value of the positive ratio

obtained in the learning process. The detailed algorithm is shown in Supplementary Fig. S12.

In learning clinical specimens of saliva, 40 PCR-positive and 40 PCR-negative specimens that were stored under refrigerated condition have been used. The number of nanopore modules used and the corresponding waveforms obtained are shown in Supplementary Table S5. The accuracy of discrimination between PCR-positive and PCR-negative by one ion current–time waveform is obtained as $F = 1.00$. When this value is >0.50, it is possible to distinguish between the positive and negative specimens with one waveform. The sensitivity and specificity at the measurement time of $n$ minutes ($n = 1$–5) are calculated using Equations 1–4 shown in Supplementary Fig. S5 based on the confusion matrix (Supplementary Fig. S13) at each measurement time. The sensitivity and specificity are time independent and are both 100% (Fig. 3f). In the true positive sample, all waveforms exhibited high positive confidence during the measurement time, while in the true negative sample, all waveforms displayed low positive confidence (Supplementary Fig. S14). The errata for all specimens obtained from the 5 min measurement along with the positive confidence and ratio are given in Supplementary Data 1.

Using the classifier obtained in the learning process, 50 PCR-positive samples and 50 PCR-negative samples are diagnosed independently of the training data. The number of used nanopore modules and the corresponding obtained waveforms are shown in Supplementary Table S6. The sensitivity and specificity at $n$ minutes ($n = 1$–5) are calculated based on the confusion matrix (Supplementary Fig. S15). Both sensitivity and specificity increased with an increase in the measurement time (Fig. 3g). After 5 min, the sensitivity and specificity are established to be 90% and 96%, respectively. Thus, the sensitivity and specificity are lower than those in the learning process, which could be attributed to overfitting. The errata for all specimens obtained from the 5 min measurement along with the positive confidence and ratio are shown in Supplementary Data 2. Figure 3h–k demonstrates the time dependence of the waveform confidence during a 5-min measurement. In the scatter plot, the orange and green points correspond to the waveform, which is determined to be positive and negative, respectively. The time dependence of the confidence of the waveform in the true positive sample suggests that the waveform with a high positive confidence is continuously large during the measurement and that the positive waveform ratio is also large (Fig. 3h). Conversely, the dependence of the waveform confidence in the true negative sample indicates that the number of waveforms with high positive confidence is small and that the positive waveform ratio is also small (Fig. 3i). Among false positive specimens, there are waveforms with a high positive confidence and a relatively large positive waveform ratio (Fig. 3j). Conversely, false negative specimens exhibit a smaller number of waveforms with a high positive confidence and a smaller positive waveform ratio (Fig. 3k). The sensitivity of 90% gives a false negative rate of 10%; therefore, it could be used for screening tests aimed at diagnosing a large number of people with a high throughput.

## Discussion

This measurement platform using machine learning consists of nanopore devices, measuring devices, and machine learning. Since nanopore devices are manufactured within the processing accuracy (diameter error ±10 nm), there is no similar platform. The measurements are not made on the same platform in this case, and it is expected that the nanopore device could affect the discriminant $F$-value. That is, machine learning might be able

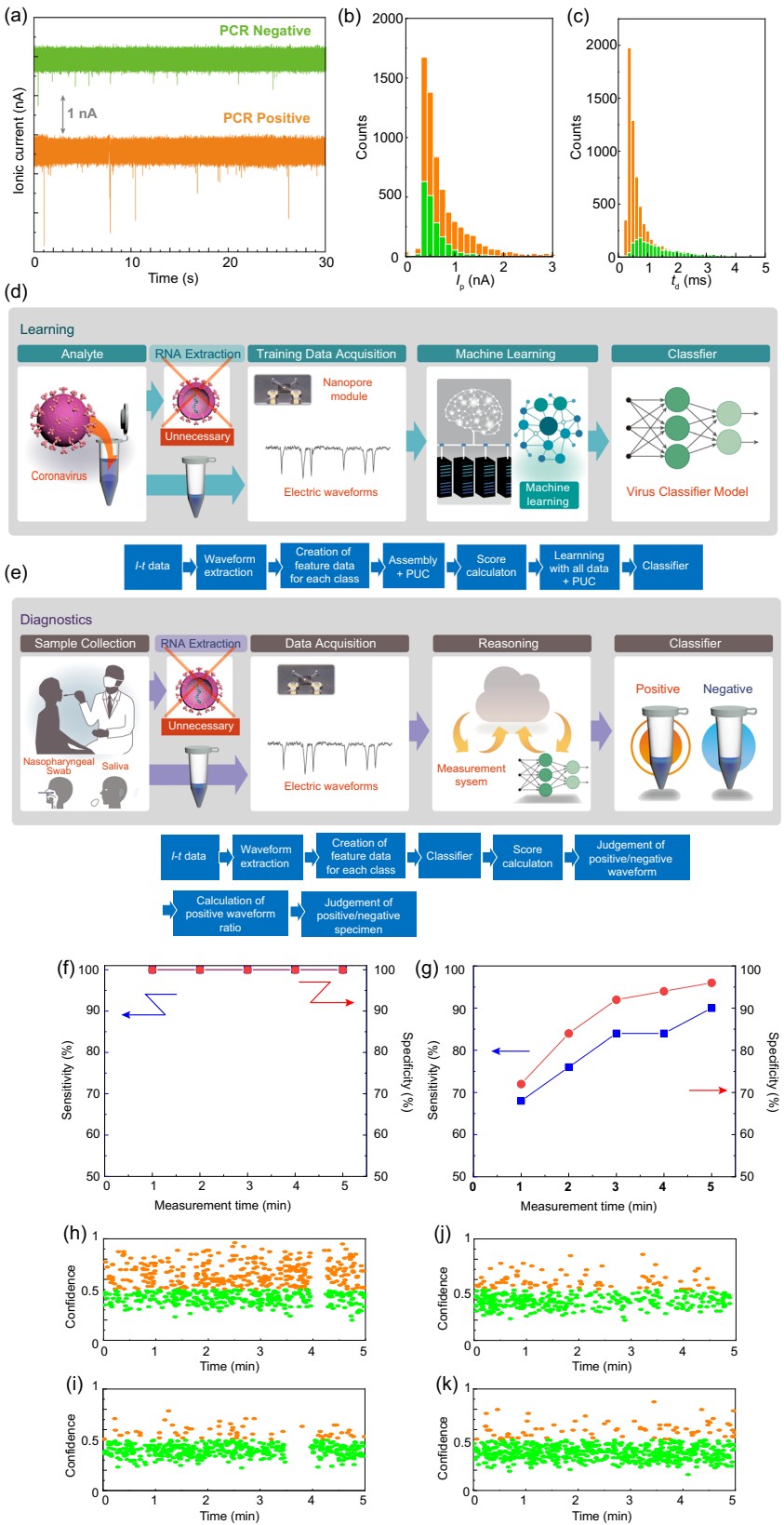

to distinguish between nanopore devices rather than nanoparticles and viruses. To show that machine learning does not differentiate between nanopore devices, we can use the same nanopores to measure the combination of SARS-CoV-2, SARS-CoV, MERS-CoV, and HCoV-229E. However, biosafety level 3 (BSL-3) facilities can handle SARS-CoV and SARS-CoV-2, while BSL-3 facilities can handle MERS-CoV, and BSL-2 facilities can handle HCoV-229E. From the viewpoint of preventing contamination by other viruses, it is impossible to conduct experiments that simultaneously handle these viruses. However, the $F$-values $= 0.96$–$1.0$ (Supplementary Fig. S6) that discriminates between nanoparticles with diameters of 200 and 220 nm using

**Fig. 3 Learning and diagnosis of clinical specimens.** Both processes do not require RNA extraction from the virus. **a** Ionic current–time traces of the PCR-positive and PCR-negative saliva specimens. **b**, **c** Histograms of $I_p$ and $t_d$ of saliva specimens. The color code corresponds to **a**. $I_p$ and $t_d$ show the peak current and duration of the current of an ionic current waveform, respectively. **d** Learning process of PCR-positive and PCR-negative saliva specimens using the learning algorithm. In all, 40 PCR-positive and 40 PCR-negative saliva samples were used for learning. **e** Diagnosis process of PCR-positive and PCR-negative saliva specimens utilizing the learning algorithm. In all, 50 PCR-positive and 50 PCR-negative saliva samples were used for learning. Time-dependent sensitivity and specificity of saliva specimens in **f** the learning process and **g** diagnostic process. The blue and red dots indicate sensitivity and specificity, respectively. Time dependence of the waveform confidence for clinical saliva specimens in the diagnostic process that showed **h** true positive (#4378), **i** true negative (#4423), **j** false positive (#4420), and **k** false negative (#4385), respectively. Sample identification numbers correspond to the errata in Supplementary Data 2. Orange and green circles indicate the positive and negative waveforms, respectively. Source data of **a–c** and **f–k** are provided as a Source Data file.

different nanopore devices are identical to the $F$-values $=$ 0.96–0.99 (Supplementary Fig. S7) used to distinguish between these two nanoparticles using the same nanopore devices. As a result, the difference between the nanopore devices has no effect on the F-value within the range of the processing accuracy of the nanopore device used in this study.

The artificial intelligent nanopore was successful in the detection of virus with high accuracy. By changing the training data from cultured viruses to PCR-positive/-negative specimens, the artificial intelligent nanopore platform becomes a device capable of detecting both positive and negative specimen with high sensitivity at high throughput. By modifying the training data, the platform is a versatile virus diagnostic system. For instance, infections caused by influenza A virus, which usually spreads between autumn–winter every year, will show the similar symptoms as caused by SARS-CoV-2[33–35]. When a person infected by the new coronavirus, based on the flu-like symptoms approaches for medical aid, the risk of infection to medical staffs and the spreads the infection increases if not diagnosed accurately. According to this study, machine learning of the cultured SARS-CoV-2 and influenza A virus (H1N1) showed an extremely high discriminator with an $F$-value of 0.90 (Supplementary Fig. S16). When the clinical specimens collected from patients infected with each virus are used as training data, the identification result of the cultured viruses will enable the development of a device that can diagnose both viruses with high accuracy.

## Methods

**Preparation of cultured viruses.** African green monkey kidney Vero cells (ATCC®CCL-81™) and human cervix adenocarcinoma HeLa cells (ATCC) were maintained and grown in DMEM (Nacalai Tesque, Kyoto, Japan) containing 5% fetal bovine serum (Thermo Fisher Scientific, MA, USA). SARS-CoV Frankfurt strain, MERS-CoV/EMC2012 strain, and SARS-CoV-2/Hu/DP/Kng/19-020 strain (GenBank Accession number LC528232) were propagated using Vero cells. HCoV-229E (ATCC VR740) was propagated in HeLa cells. Viral stocks of these coronaviruses were filtered using Millex-HV Syringe Filter Unit 0.45 µm (Merck, Darmstadt, Germany) and the filtrates were used for diagnostic method using nanopore and machine learning. Viral titers were calculated by the modified 50% tissue culture infectious dose (TCID$_{50}$) assay and plaque assay[36,37]. The genome copy number of HCoV-229E was determined by quantitative PCR using Taqman probe. Influenza A (H1N1pdm09, California/7/2009) virus was added to MDCK cells (ECACC84121903) in DMEM, and after incubation for 6 h, trypsin was added at a final concentration of 2.5 µg/ml. The infected cells were incubated until a cytopathic effect was observed. The medium of the infected cells was centrifuged at 440 × g for 5 min, the supernatant of medium was collected, and filtered with a filter having a pore size of 0.45 µm (Millex-HV; Millipore Co.). All the experiments using influenza virus were approved by the institutional biosafety committee, and precautiously carried out in BSL-2 facilities.

**Clinical specimens.** Specimens were acquired as residual samples from clinical examination with the approval of Institutional Review Board (IRB), Osaka University Hospital, Osaka University, Japan. The need to obtain informed consent was waived by the IRB committee since the samples were residual from a clinical examination without using any identifiable information of the individuals or the application of any intervention, in accordance with the Ethical Guidelines for Medical and Health Research Involving Human Subjects. (Public Notice of the Ministry of Education, Culture, Sports, Science and Technology and the Ministry of Health, Labor and Welfare No. 3 of 2014).

**Artificial intelligence nanopore platform.** A nanopore module, nanopore measuring instrument, and Aipore-ONE™, artificial intelligence software developed and implemented by Aipore Inc. were used. HCoV-229E was measured at the BSL-2 facility. SARS-CoV, SARS-CoV-2, and MERS-CoV were measured at the BSL-3 facility at the Research Institute for Microbial Diseases, Osaka University. The experiments were confirmed by the Institutional Biosafety Committee. The nanoSCOUTER™ was used for all measurements. The bias voltage was 0.1 V. The current amplification factor was $10^7$, and it had the characteristic of $F_c = 260$ KHz and the sampling rate was set as 1 MHz.

**PCR analysis.** Viral RNA of HCoV-229E was purified using TRIzol LS Reagent (Thermo Fisher Scientific). Viral RNA copy numbers were quantified using QuantiTect Probe RT-PCR Kit (Qiagen GmbH, Hilden, Germany) and PCR thermal cycler Dice (Takara). Primers and probes (Eurofins Genomics, Ebersberg, Germany) specific to the viral RNA-dependent RNA polymerase (RdRp) are listed in Supplementary Table S7. For standard of genome copy number, the partial nucleotides of RdRp (position 998-1999 of HCoV-229E, GenBank Accession number MT438700) was amplified using forward and reverse primers listed in Supplementary Table S7 and cloned into pCAGGS plasmid using Gibson assembly kit (New England Biolabs, Germany).

Saliva samples were examined by SARS-CoV-2 Direct Detection RT-qPCR Kit (Takara Bio, Otsu, Shiga, Japan) using Roche Light Cycler 96 for the detection of SARS-CoV-2 according to the manufacturer instructions. Primers and probes recommended by the CDC for N1, N2, and RNase P targets in a multiplex reaction are listed in Supplementary Table S7. In SARS-CoV-2 positive samples, viral copy numbers were calculated based on the $C_t$ value of RT-qPCR.

## Data availability

Source data are provided with this paper. The measurement data of the current–time profile of the nanoparticles, four types of viruses, and clinical specimens obtained in this study are available on Zenodo (https://doi.org/10.5281/zenodo.4761714). We provide the temporary account that enables researchers to login the Aipore server through a web browser to reproduce the AI model on Aipore-One™, deposited in Zenodo (https://doi.org/10.5281/zenodo.4761714) as a README file. Source Data are provided with this paper.

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

## Acknowledgements

This research was supported by AMED under the Grant Number JP20he0722002. We thank Dr Bart L. Haagmens (Erasmus Medical Center) for providing the MERS-CoV/EMC2012 strain through Dr Makoto Takeda (National Institute of Infectious Diseases), Dr John Ziebuhr (University of Würzburg) for providing the Frankfurt strain of SARS-CoV through Dr Fumihiro Taguchi, and Dr Tomohiko Takasaki (Kanagawa Prefectural Institute of Public Health) for providing the SARS-CoV-2/Hu/DP/Kng/19-020 strain.

## Author contributions

M. T. conceived the technology, supervised the project, and wrote the manuscript with input from co-authors. A. M., S. H., Y. A., Y. Y., K. S., N. H., and K. T. prepared clinical specimens and supervised the clinical specimen experiments. S. M., R. H., C. O., Y. K., T. K., W. K., Y. T., and Y. M. cultivated coronavirus and conducted experiments in BSL-3. K. T. cultured influenza virus. N. W. has developed a nanopore instrument. H. T. and N. N. conducted experiments on BSL-2 and measured clinical specimens. O. S. and T. W. developed the software for machine learning.

## Competing interests

M. T. and N. N. are the co-founders of Aipore Co., Ltd., and its Director and Chief Executive Officer, respectively. The authors other than M. T. and N. N. have no competing interests to declare.
