## [Peer Review File · Nature Communications]

Reviewers' Comments:

Reviewer #1:

Remarks to the Author:

The manuscript by Taniguchi et al. reports the use of nanopore sensors and an artificial intelligence classifier to directly measure the presence of SARS-CoV-2 virus particles in liquid samples.

The most striking impression this manuscript leaves is that it reads like promotional material, rather than a technical research report. The authors make strong claims about the speed, sensitivity, and especially the selectivity of the technique. Unfortunately, the manuscript contains very little detail about the methods or the results. The figures have a high production value and are visually appealing, but they contain very little information. Overall, I was not convinced of the principal claims by the evidence presented.

The number of events used in the learning process – between 100 and 1000 – is quite low by the standards of nanopore measurements. What was the precise number of events used to train the classifier for each type of virus? How many nanopores in total were tested, and was it necessary to train the classifier with each one separately? The authors should make their raw data publicly available through one of the standard platforms, like the Harvard Dataverse, to help ensure that the analyses are transparent and reproducible. The manuscript itself should also present a representative selection of the training set data. It is important to know what feature or features of the current blockades the classifier is recognizing. It is possible that SARS-CoV-2 particles can be accurately recognized based on its size and shape, but it is also possible, for example, that the classifier spuriously recognizes the distinct electrical noise produced by a researcher hovering over the electrometer.

The authors claim that “During the diagnostic process, it is observed that the prepared classifier is successful in determining the positive/negative nature of the clinical specimen in a duration of 5 minutes (Figure 2b)”. However, Fig. 2b presents no real evidence to support that claim.

I am also concerned by other statements the authors made in reporting the detection of virus particles. First, the rates at which the authors reportedly measured virus particles are minuscule; the authors appear to be claiming to identify strains of virus on the basis of only a handful of detection events. They should report the actual number of events and present the raw current traces. If virus identification is possible with so few events and with the accuracy the authors claim, then the individual current traces are likely distinguishable by eye. It is also concerning that the same type of virus particle would be observed translocating the nanopore using a driving voltage with either polarity.

The authors evaluate the performance of their diagnostic system by measuring nanoparticles with diameters of 200 nm and 210 nm. The 15% volume difference between those particles is expected to be relatively easy to detect. Is a similar volume difference the basis for discriminating the various virus particles studied here? The authors claim that their classifier was unaffected by manufacturing variations between different nanopores; they should present raw and analyzed results to support this claim.

Finally, the authors need to cite the literature properly. Every claim about nanopores refers back to a single recent review paper by one of this paper’s authors, who is not a recognized authority in the field. The authors portray the direct detection of viruses as particularly challenging because the ionic current signal is “very feeble”. But nanopores have already been used to directly detect single virus particles, as was reported by McMullen et al. in Nature Communications in 2014.

Reviewer #2:

Remarks to the Author:

In this work, Taniguchi and co-worker report a nanopore platform combined with artificial intelligence for the detection of SARS-CoV-2. The manuscript is on an important topic and interesting. However, it suffers from significant flaws based on the lack of essential data. I cannot recommend this manuscript for publication in nature communication.

1. There is no current trace
2. The histogram or event map for all samples are not reported
3. The different event parameters obtained from different viruses are not discussed
4. What is the real input of the learning machine compared to classical clustering or PCA
5. There is no information about the algorithm used for the learning machine analysis
6. The work combined machine learning and nanopore are not cited.

Reviewer #3:

None

Reviewer #4:

Remarks to the Author:

An artificial intelligence nanopore platform for SARS-CoV-2 virus detection

Masuteru Taniguchi and others,

In this manuscript the authors present a nanopore-based technology that they seek to apply to the detection of coronaviruses. The fundamental idea supporting the work is the creation of a microfluidic nanopore setup where current can be measured as samples are introduced into the microchamber and electricity is introduced to encourage particle movement through a nanopore. The authors present the idea that the nanopores can "detect" the presence of viruses presumably passing through the pores as a transient change in current that is unique to each virus tested. If true, this could enable a transformative diagnostic approach that does not need significant sample preparations and can work quickly. While this idea is very captivating, this reviewer's enthusiasm for this manuscript was diminished on a number of fronts. There is very little primary data presentation in this manuscript. Two of the four figures are cartoon schematics that summarize what are very important steps in the whole process, and this makes it difficult to analyze the rigor of the underlying science. A second, related factor diminishing enthusiasm was the lack of understanding of how the learning algorithms functioned and how many samples were used in the training set vs. how many were used in the testing set. A third factor diminishing enthusiasm was that there was not enough data presented to understand how the positive/negative calls were made on clinical samples and what the overall level of virus was in these samples. Sensitivity and specificity numbers can vary greatly depending on viral load in the samples used in these very small number tests. There were minor concerns with the writing.

Specific major concerns.

- 1) The construction of the devices is confusing to this reviewer, as the authors reference a 300nM nanopore and then suggest that it is effective at discriminating viruses with a diameter of half that size and polystyrene beads $\sim 2/3$ of the 300nM value. This may be due to a lack of data presented.
- 2) The use of schematics to illustrate the most important aspects of the underlying science is problematic for understanding the rigor of the experiments. The primary data should be shown.
- 3) Supplemental figure 1 shows "typical" traces of solutions with virus present. These traces appear to be quite noisy and it is difficult to appreciate what is different about each of the traces and how much variation there is in different traces from the same virus. Presentation of multiple traces overlaid, as

well as averages would improve the understanding of the rigor of measurement.

4) The factors that go into the performance testing of the classifier are very difficult for this reviewer to appreciate. It is not clear whether one classifier is built for each sample or if there is a single classifier for each sample type. Testing and training data are not well differentiated in the description.

5) A confusing aspect to this reviewer is that the solutions that the authors test are known to contain a very large number of other nano and microparticles, ranging from extracellular vesicles to microvesicles. The authors note that transit through their nanopores is non-specific (current based) and so these vesicles should also move through or block pores. It is unclear how these particles contribute to the overall signal seen and whether they can occlude or influence signal from the much lower abundance viruses. In addition, NP swab samples are cloudy because of the nature of the sample having nano to milli-scale particles. This seems to be a difficult medium for a nanopore setup to work in without clogging. The lack of primary data hampers an understanding of nanopore function under different conditions.

6) It is confusing to this reviewer how the authors see the channels providing specificity. It would be helpful to provide an explanation of how they differentiate virus particles from other nanoparticles. The coronaviruses are essentially the same size so that can't be the distinguishing factor in their "learning" model.

7) The clinical sample analysis does not provide enough understanding of what samples were used for training and what samples were used for testing. How were sensitivity and specificity determined? In the presented results, there is little change in specificity and sensitivity over time, which seems to run counter to the overall idea of the technology, where waveform counting should increase over time.

Minor Comments

Throughout the manuscript the writing style leads to confusion. This is likely something that could be fixed with careful editing.

There are some claims made in the manuscript that should likely be softened. These include:

"Here we show that the artificial intelligent nanopores are successful in accurate identification of four types of coronaviruses, HCoV-229E, SARS-CoV, MERS-CoV, and SARS-CoV-2, which are usually extremely difficult to detect."

The suggestion that coronaviruses are difficult to detect is inaccurate. Powerful and effective for all of these viruses exist already and portable point of care assays for these SARS-CoV-2 already exist. SARS-CoV-2 is diagnosed about a million times a day at the current status of the outbreak. The authors should address a different aspect of the advantages of their technology.

"The platform enables high throughput diagnostics with low false negatives for the novel coronavirus"

While the potential for the technology is appreciated, the authors have not demonstrated high-throughput diagnostics or carried out a thorough false-negative analysis. The claims should be significantly modified to reflect this.

"However, the RT-PCR method is prone to false negative determination which increases the risk of viral infection."

This is an unreferenced statement that does not reflect the current state of coronavirus diagnostics. The manuscript should be modified to accurately reflect the continuing, major, and successful role of PCR and other nucleic acid technologies in modern coronavirus diagnostics.

.....
An artificial intelligence nanopore platform
for SARS-CoV-2 virus detection
.....

By Masateru Taniguchi, Shohei Minami, Chikako Ono, Rina Hamajima, Ayumi Morimura, Shigeto Hamaguchi, Yukihiro Akeda, Yuta Kanai, Takeshi Kobayashi, Wataru Kamitani, Yutaka Terada, Koichiro Suzuki, Nobuaki Hatori, Yoshiaki Yamagishi, Nobuei Washizu, Hiroyasu Takei, Osamu Sakamoto, Norihiko Naono, Kenji Tatematsu, Takashi Washio, Yoshiharu Matsuura & Kazunori Tomono

We would like to express our sincere gratitude to the referees for their valuable and helpful comments. We have studied the comments carefully and have incorporated the following revisions.

We concur with the unanimous comment provided by all reviewers on the lack of sufficient data for this study during initial submission. We have attempted to completely address the shortcomings and the manuscript has been extensively revised. The following data and discussions have been added in the main text and supporting information.

- List of additional content included in the revised manuscript:
The following attributes are included in the main text:
Figure 1f. Raw ionic current-time traces of nanoparticles with diameters of 200 nm and 220 nm.
Figure 1g. Ionic current-time profiles which overlay 100 ionic current-time waveforms of 200 nm-nanoparticle.
Figure 1f. Ionic current-time profiles which overlay 100 ionic current-time waveforms of 220 nm-nanoparticle.
Figure 1i. Histograms corresponding to I_p of the nanoparticles.
Figure 1j. Histograms pertaining to t_d of the nanoparticles.
Figure 1k. Machine learning algorithm to identify the nanoparticles and cultured viruses.
Figure 1l. Confusion matrix of nanoparticles.
Figure 2a. Raw ionic current-time traces of four cultured coronaviruses.
Figure 2b. Ionic current-time profiles which overlay 100 ionic current-time waveforms

of 4 cultured coronaviruses.
Figure 2c. Histograms corresponding to I_p for four cultured coronaviruses.
Figure 2d. Histograms representing the t_d of four cultured coronaviruses.
Figure 2f. Confusion matrix of four cultured coronaviruses.
Figure 3a. Raw ionic current-time traces of PCR-positive and PCR-negative specimens of
nasopharyngeal swab.
Figure 3b. Raw ionic current-time traces of PCR-positive and PCR-negative specimens
of saliva.
Figure 3c. Histograms corresponding to I_p of PCR-positive and PCR-negative specimens
of nasopharyngeal swab.
Figure 3d. Histograms pertaining to t_d of PCR-positive and PCR-negative specimens of
nasopharyngeal swab.
Figure 3e. Histograms of I_p of PCR-positive and PCR-negative specimens of saliva.
Figure 3f. Histograms of t_d of PCR-positive and PCR-negative specimens of saliva.
Figure 4b. Machine learning algorithm for learning process.
Figure 4d. Machine learning algorithm for diagnostic process.
Figure 4e and 4f. Time dependences of confidence of waveforms in PCR-positive and
PCR-negative specimens of nasopharyngeal swab which were accurately predicted.
Figure 4h and 4i Time dependences of confidence of waveforms in PCR-positive and
PCR-negative specimens of saliva which were accurately predicted.
Additional information included in the supporting information are:
Table S1. Nanopore modules used to measure nanoparticles.
Table S2. Details on waveforms obtained from nanoparticle measurements and that
utilized for machine learning.
Table S3. Number of waveforms obtained from measurements of four corona viruses
and number of waveforms used for machine learning.
Table S4. Waveform data obtained from measurements of nasopharyngeal swab and
that used for machine learning.
Table S5. Number of waveforms collected from measurements of saliva and number of
waveforms used for machine learning.
Table S6. Errata for diagnosis obtained by machine learning a 5-minute measurement
of nasopharyngeal swab.
Table S7. Errata for diagnosis obtained by machine learning a 5-minute measurement

of saliva.
Figure S1. Machine learning details of nanoparticles and cultured coronaviruses.
Figure S2. Under sampling method.
Figure S3. 10-Fold cross validation method.
Figure S4. Confusion matrix calculated by machine learning.
Figure S5. Confusion matrixes of nanoparticles with diameter of 200 nm and 220 nm.
Figure S8. Detailed algorithm of learning process of clinical specimens.
Figure S9. Detailed algorithm of diagnosis process of clinical specimens.
Figure S10. Time dependences of confidence of waveforms in PCR-positive and
PCR-negative specimens of nasopharyngeal swabs which were inaccurately predicted.
Figure S11. Time dependence of confusion matrices of nasopharyngeal swabs.
Figure S12. Time dependences of confidence of waveforms in PCR-positive and
PCR-negative specimens of saliva which were not accurately determined.
Figure S13. Time dependence of confusion matrices of saliva.

**Reviewers' comments:**

**Reviewer #1 (Remarks to the Author):**

**The most striking impression this manuscript leaves is that it reads like promotional**
**material, rather than a technical research report. The authors make strong claims**
**about the speed, sensitivity, and especially the selectivity of the technique.**
**Unfortunately, the manuscript contains very little detail about the methods or the**
**results. The figures have a high production value and are visually appealing, but they**
**contain very little information. Overall, I was not convinced of the principal claims by**
**the evidence presented.**

**Response**

**As the reviewer has indicated, we understand the immediate requirement to include**
**more data to support our claims. We have incorporated the changes in the revised**
**manuscript in order to improve its significance. The additional content in the revised**
**manuscript is as listed in the previous section.**

**Comment**

**The number of events used in the learning process – between 100 and 1000 – is quite**
**low by the standards of nanopore measurements. What was the precise number of**

events used to train the classifier for each type of virus? How many nanopores in total
were tested, and was it necessary to train the classifier with each one separately? The
authors should make their raw data publicly available through one of the standard
platforms, like the Harvard Dataverse, to help ensure that the analyses are transparent
and reproducible. The manuscript itself should also present a representative selection
of the training set data. It is important to know what feature or features of the current
blockades the classifier is recognizing. It is possible that SARS-CoV-2 particles can be
accurately recognized based on its size and shape, but it is also possible, for example,
that the classifier spuriously recognizes the distinct electrical noise produced by a
researcher hovering over the electrometer.

Response

We have revised the misleading expressions and have added the number of nanopore
modules used for measurement, the number of waveforms obtained, and the number
of waveforms used for machine learning. The modifications are included in the Tables
S1-S5 of Supplementary Information. The features used in the machine learning
algorithm are specified in the text.

As the reviewer has stated, electrical signals which do not correspond to SARS-CoV-2
are treated as noise. In this paper, the signal obtained by measuring PCR-negative
samples is assumed to be a noise signal which is not caused by SARS-CoV-2. We also
assume that the signals obtained from PCR-positive samples are due to both
SARS-CoV-2 and noise. Noise is learned from learning the signals of PCR-negative
samples. Subsequently, the signal of the PCR-positive sample is learned, and the
learned noise signal is removed from the signal of the PCR-positive sample. This is the
principle of a classifier called Positive Unlabeled Classifier (PUC). In order to clearly
understand this algorithm, the corresponding flow diagram is shown in Figures 4b and
4d in the main text and Figures. S8 and S9 in Supplementary Information. In addition,
an appropriate explanation was also added to the main text.

Comment

The authors claim that “During the diagnostic process, it is observed that the prepared
classifier is successful in determining the positive/negative nature of the clinical
specimen in a duration of 5 minutes (Figure 2b)”. However, Fig. 2b presents no real
evidence to support that claim.

Response

In accordance with the reviewer's comment, the sequence of presenting the figures in
the original manuscript was changed to avoid confusion. The 5 minutes shown in the
figure are the results obtained in this study and are inappropriate to be included in the
figure. In this regard, we have deleted the 5-minute description and revised the figure.
Also, to avoid misleading the readers, the highlighted sentence has been deleted.

Comment

I am also concerned by other statements the authors made in reporting the detection
of virus particles. First, the rates at which the authors reportedly measured virus
particles are minuscule; the authors appear to be claiming to identify strains of virus on
the basis of only a handful of detection events. They should report the actual number
of events and present the raw current traces. If virus identification is possible with so
few events and with the accuracy the authors claim, then the individual current traces
are likely distinguishable by eye. It is also concerning that the same type of virus
particle would be observed translocating the nanopore using a driving voltage with
either polarity.

Response

We have added information pertaining to the raw ionic current-time traces to Figure 2a
of the main text. The number of nanopore modules used, the number of waveforms
obtained, and the number of waveforms used for machine learning were added to
Table S3 of Supplementary Information.

Comment

The authors evaluate the performance of their diagnostic system by measuring
nanoparticles with diameters of 200 nm and 210 nm. The 15% volume difference
between those particles is expected to be relatively easy to detect. Is a similar volume
difference the basis for discriminating the various virus particles studied here? The
authors claim that their classifier was unaffected by manufacturing variations between
different nanopores; they should present raw and analyzed results to support this
claim.

Response

Nanoparticle measurement and machine learning algorithm reveal the performance of
the developed nanopore platform. However, as the reviewer had indicated, the particle
size of nanoparticles must be precisely confirmed in order to demonstrate the

performance. Since the 200 nm and 210 nm nanoparticles were not standard
nanoparticles, we have repeated the procedure by using NIST (National Institute of
Standards and Technology)-approved standard nanoparticles with 200 nm and 220 nm
diameters. The particle size of these nanoparticles is ± 6 nm. We have included raw
ionic current-time traces of 200 nm and 220 nm nanoparticles (Figure 1f), standard
histograms (Figure 1i and 1j), and confusion matrixes obtained by machine learning
(Figure 1l) to the main text of the revised manuscript.

Measurements were conducted using three nanopores designed with a diameter of
300 nm. The identification accuracy was compared using the data obtained from the
nanoparticles (A, B, C) measuring 200 nm nanoparticles and the nanopores (D, E, F)
measuring 220 nm nanoparticles. When the identification accuracy was calculated for
all combinations, (A, D), (A, E), (A, F), (B, D), (B, E), (B, F), (C, D), (C, E), (C, F), it was 97%
to 100%. This indicates that the identification accuracy is high and within the reliable
limits. The confusion matrices which support these results have been added to Table S5
of Supplementary Information.

Comment

Finally, the authors need to cite the literature properly. Every claim about nanopores
refers back to a single recent review paper by one of this paper's authors, who is not a
recognized authority in the field. The authors portray the direct detection of viruses as
particularly challenging because the ionic current signal is "very feeble". But nanopores
have already been used to directly detect single virus particles, as was reported by
McMullen et al. in Nature Communications in 2014.

Response

In accordance with the reviewer's suggestion, we have re-examined the literature
carefully and have cited the original papers.

Reviewer #2

**In this work, Taniguchi and co-worker report a nanopore platform combined with**
**artificial intelligence for the detection of SARS-CoV-2. The manuscript is on an**
**important topic and interesting. However, it suffers from significant flaws based on**
**the lack of essential data. I cannot recommend this manuscript for publication in**
**nature communication.**

Response

As the reviewer has indicated, we realize that the data provided was not sufficient. In
the revised manuscript, we have added data pertaining to different sections of the
manuscript as listed previously.

Comment

There is no current trace.

Response

We have added current traces in the Fig. 1a for nanoparticles, Fig. 2a for 4 cultured
coronaviruses, and Fig. 3a for clinical specimens of the main text.

Comment

The histogram or event map for all samples are not reported.

Response

Histograms of peak current values and current durations, Figs. 1i and 1j for
nanoparticles, Figs. 2c and 2d for 4 cultured coronaviruses, and Figs. 3c-3f clinical
specimens, have been added to the main text.

Comment

The different event parameters obtained from different viruses are not discussed.

Response

We have included a segment in the main text which discusses the peak current (I_p) and
current duration (t_d) based on histograms of I_p and t_d of the four cultured
coronaviruses.

Comment

What is the real input of the learning machine compared to classical clustering or PCA.

Response

The features used in machine learning are peak current values (I_p), current durations
(t_d), current vectors, time vectors, and features that combine these features (eg, $I_p + t_d$).

The current vector (I_1, I_2, \dots, I_{10}) and time vector (t_1, t_2, \dots, t_{10}) are obtained by
dividing the waveform data into 10-folds along the time and current directions. In

order to obtain one waveform information, the current vector and the time vector
were used instead of the scalar quantities I_p and t_d . The above content has been added
to the main text.

Comment

There is no information about the algorithm used for the learning machine analysis.

Response

The machine learning algorithms for nanoparticles and cultured coronaviruses, and the
machine learning algorithms for clinical specimens have been added to Figs 1k and
S1-S4, and Figs. 4b, 4d, S8, and S9, respectively.

Comment

The work combined machine learning and nanopore are not cited.

Response

In accordance with the reviewer's suggestion, we have re-examined the literature
carefully and have cited relevant articles.

**Reviewer #3 (Expertise: Virology):**

**In this manuscript the authors present a nanopore-based technology that they seek**
**to apply to the detection of coronaviruses. The fundamental idea supporting the**
**work is the creation of a microfluidic nanopore setup where current can be measured**
**as samples are introduced into the microchamber and electricity is introduced to**
**encourage particle movement through a nanopore. The authors present the idea that**
**the nanopores can "detect" the presence of viruses presumably passing through the**
**pores as a transient change in current that is unique to each virus tested. If true, this**
**could enable a transformative diagnostic approach that does not need significant**
**sample preparations and can work quickly. While this idea is very captivating, this**
**reviewer's enthusiasm for this manuscript was diminished on a number of fronts.**
**There is very little primary data presentation in this manuscript. Two of the four**
**figures are cartoon schematics that summarize what are very important steps in the**
**whole process, and this makes it difficult to analyze the rigor of the underlying**
**science. A second, related factor diminishing enthusiasm was the lack of**
**understanding of how the learning algorithms functioned and how many samples**
**were used in the training set vs. how many were used in the testing set. A third factor**

**diminishing enthusiasm was that there was not enough data presented to**
**understand how the positive/negative calls were made on clinical samples and what**
**the overall level of virus was in these samples. Sensitivity and specificity numbers**
**can vary greatly depending on viral load in the samples used in these very small**
**number tests. There were minor concerns with the writing.**

**Response**

**As the reviewer has indicated, we understand the serious lack of data provided during**
**initial submission. In the revised paper, we have added relevant information as listed**
**previously.**

**Specific major concerns.**

**Comment**

**The construction of the devices is confusing to this reviewer, as the authors reference a**
**300nM nanopore and then suggest that it is effective at discriminating viruses with a**
**diameter of half that size and polystyrene beads ~2/3rds of the 300nM value. This may**
**be due to a lack of data presented.**

**Response**

**In order to address the reviewer's comment, raw ionic current-time data of polystyrene**
**nanoparticles and four coronaviruses have been added to Figure 1a and Figure 2a of**
**the main text, respectively. In addition, the confusion matrices of two nanoparticles**
**and the confusion matrices of four coronaviruses using machine learning are shown in**
**Figure 1l of the main text and Figure S5 of Supplementary Information, and Figure 2f of**
**the main text and Figures S6 and S7 of Supplementary Information, respectively.**

**Comment**

**The use of schematics to illustrate the most important aspects of the underlying**
**science is problematic for understanding the rigor of the experiments. The primary**
**data should be shown.**

**Response**

**In order to provide evidence for the rigor and significance of the study, raw ionic**
**current-time traces of clinical specimens have been added to Figures 3a and 3b of the**
**main text.**

Comment

Supplemental figure 1 shows “typical” traces of solutions with virus present. These
traces appear to be quite noisy and it is difficult to appreciate what is different about
each of the traces and how much variation there is in different traces from the same
virus. Presentation of multiple traces overlaid, as well as averages would improve the
understanding of the rigor of measurement.

Response

In order to provide emphasis to the rigor of measurement in this study and to address
the reviewer’s comment the following modifications are incorporated in the
manuscript. Figures in which 100 current-time waveforms of each of the four cultured
viruses are overwritten and have been added to Figure 2b of the main text. In addition,
histograms of the peak current values and current durations of ionic current-time
waveforms of the four cultured coronaviruses have been added to Figures 2c and 2d of
the main text. Similarly, for the nanoparticles, 100 overlaid ionic current-time
waveforms have been added to Figures 1g and 1h of the main text.

Comment

The factors that go into the performance testing of the classifier are very difficult for
this reviewer to appreciate. It is not clear whether one classifier is built for each sample
or if there is a single classifier for each sample type. Testing and training data are not
well differentiated in the description.

Response

In order to provide clarity on the machine learning algorithms used for the clinical
specimen, we have added the corresponding algorithms to Figures 4b and 4d of the
main text and Figures S8 and S9 of the Supplementary Information to clarify the
contents of machine learning of clinical specimens. During the learning stage, many
classifiers are compared, and the one that yields the highest accuracy (*F*-value) is used
for diagnosis. In this study, the classifier utilized for diagnosis varies with the material
or virus that is being detected.

In the learning and diagnosis stages of clinical specimens of nasopharyngeal swabs,
clinical specimen constituted by refrigerated samples of 15 PCR-positive and 26
PCR-negative specimens have been used. Of the 41 samples, 40 are used for learning
and 1 is used for testing (diagnosis). For example, 14 PCR-positive and 26 PCR-negative

specimens are used for learning and 1 PCR-positive specimen is used for diagnosis. The
method of using one specimen for diagnosis is considered for all combinations of
specimens.

Comment

A confusing aspect to this reviewer is that the solutions that the authors test are
known to contain a very large number of other nano and microparticles, ranging from
extracellular vesicles to microvesicles. The authors note that transit through their
nanopores is non-specific (current based) and so these vesicles should also move
through or block pores. It is unclear how these particles contribute to the overall signal
seen and whether they can occlude or influence signal from the much lower
abundance viruses. In addition, NP swab samples are cloudy because of the nature of
the sample having nano to milli-scale particles. This seems to be a difficult medium for
a nanopore setup to work in without clogging. The lack of primary data hampers an
understanding of nanopore function under different conditions.

Response

In this study, the electrical signals other than SARS-CoV-2 are considered as noise. The
nasopharyngeal swab was diluted with buffer and used for measurement. The highly
viscous saliva was filtered through a 0.45 μm membrane filter, diluted with buffer prior,
to the measurement. The counter measures pertaining to noise are taken by the
machine learning algorithm. The signal obtained by measuring the PCR-negative
specimen is assumed to be a noise signal. We also assume that the signal obtained
from the PCR-positive specimen is composed of SARS-CoV-2 and noise. Noise is learned
by learning the signals of PCR-negative specimen. Further, the signal of the
PCR-positive sample is learned, and the learned noise signal is removed from the signal
of the PCR-positive sample. This is the principle of a classifier called Positive Unlabeled
Classifier (PUC). To provide a clear understanding on this algorithm, we have added a
flow diagram in Figure 4b and Supplementary Information S8. In addition, the raw ionic
current-time traces of PCR-negative and PCR-positive specimens have been added to
Figures 3a and 3b of the main text. A description of the PUC method has also been
added to the main text.

As the reviewer has indicated, it is necessary to determine whether each waveform
corresponds to the novel coronavirus or noise. The confidence is a measure of the
accurate classification of a single waveform as positive or negative. To provide further
clarity, the time-time dependence of confidence of each waveform in clinical

specimens has been added to Figures 4e, 4f, 4h, 4i in the main text and Figures S10,
S12 in Supplementary Information. Also, a discussion section on confidence is added to
the main text and Supplementary Information.

Comment

It is confusing to this reviewer how the authors see the channels providing specificity. It
would be helpful to provide an explanation of how they differentiate virus particles
from other nanoparticles. The coronaviruses are essentially the same size so that cant
be the distinguishing factor in their "learning" model.

Response

Our research group has prior experience in studying the theoretical aspects of the flow
dynamics of nanoparticles passing through nanopores using multiphysics simulation.
As a result, we show that the ionic current-time waveform has information on the
volume, structure, and surface charge of nanoparticles. We propose that if we could
analyze the amount containing these information, we would be able to identify
nanoparticles and viruses of the same size. One of the possible methods is machine
learning, which recognizes the waveforms. The study on the ionic current-time
waveform and observing the key features such as current and time vectors, volume,
structure, and surface charge information can be used to identify nanoparticles. It is
proposed that the coronaviruses with the same volume can be identified because the
difference in coronavirus structure and surface charge is identified by machine learning.
We have modified the main text to provide clarity on the ionic current-time waveform
which provides information on the volume, structure, and surface charge of
nanoparticles passing through the nanopore. We have also included relevant literature.

Comment

The clinical sample analysis does not provide enough understanding of what samples
were used for training and what samples were used for testing. How were sensitivity
and specificity determined? In the presented results, there is little change in specificity
and sensitivity over time, which seems to run counter to the overall idea of the
technology, where waveform counting should increase over time.

Response

Machine learning algorithms have been added to Figures 4b and 4d in the main text
and Figures S8 and S9 in Supplementary Information to clarify the diagnostic method

for clinical specimens. In addition, the method of machine learning was added to the
main text. Furthermore, in order to clarify the method and time dependence of
positive / negative judgment of clinical specimens, the time-time dependence of
confidence of each waveform in clinical specimens has been added to Figures 4e, 4f, 4h,
4i in the main text and Figures S10, S12 in Supplementary Information. Also, a
discussion about confidence was added to the main text and Supplementary
Information. In addition, confusion matrixes at each time have been added to Figures
S11 and S13 of the Supplementary Information to clarify the time dependence of
sensitivity and specificity. In addition, the errata for all clinical specimens have been
added to Tables S6 and S7 of the Supplementary Information.

**Minor Comments**

**Throughout the manuscript the writing style leads to confusion. This is likely**
**something that could be fixed with careful editing.**

**Comment**

There are some claims made in the manuscript that should likely be softened. These
include:

“Here we show that the artificial intelligent nanopores are successful in accurate
identification of four types of coronaviruses, HCoV-229E, SARS-CoV, MERS-CoV, and
SARS-CoV-2, which are usually extremely difficult to detect.”

The suggestion that coronaviruses are difficult to detect is inaccurate. Powerful and
effective for all of these viruses exist already and portable point of care assays for these
SARS-CoV-2 already exist. SARS-CoV-2 is diagnosed about a million times a day at the
current status of the outbreak. The authors should address a different aspect of the
advantages of their technology.

**Response**

Based on the reviewer’s comment, we have revised the following in order to emphasize
the superiority of AI-nanopore technology.

Here we show that the artificial intelligent nanopores are successful in accurate
identification of four types of coronaviruses, HCoV-229E, SARS-CoV, MERS-CoV, and
SARS-CoV-2, which have similar sizes.

**Comment**

“The platform enables high throughput diagnostics with low false negatives for the

novel coronavirus”

While the potential for the technology is appreciated, the authors have not
demonstrated high-throughput diagnostics or carried out a thorough false-negative
analysis. The claims should be significantly modified to reflect this.

Response

The sentence has been deleted according to the reviewer’s comment.

**Comment**

“However, the RT-PCR method is prone to false negative determination which increases
the risk of viral infection.”

This is an unreferenced statement that does not reflect the current state of coronavirus
diagnostics. The manuscript should be modified to accurately reflect the continuing,
major, and successful role of PCR and other nucleic acid technologies in modern
coronavirus diagnostics.

Response

The sentence and the subsequent sentence have been deleted according to the
reviewer’s comment.

Reviewers' Comments:

Reviewer #1:

Remarks to the Author:

The inclusion of primary data has improved the revised manuscript by Taniguchi et al. Unfortunately, the raw data are still not publicly available, and this makes it impossible to ascertain the reproducibility of the results.

The revised manuscript has a number of other issues that weaken my confidence in the key claims:

1. The authors added virus the trans compartment (line 140), and later found virus in a sample drawn from the trans compartment by RT-PCR (line 143). They took this as evidence that the measured current traces were produced by virus translocations. That conclusion is unfounded.
2. The authors trained between 4 and 9 nanopores with waveforms from samples of the four types of virus. Clearly, different nanopores were used to train the classifier on different viruses. The accuracy of that classifier at identifying different virus types with the same nanopore was not reported. It is therefore possible that the classifier is in fact recognizing particular nanopores, not viruses.
3. The training procedure used to classify samples from nasal swabs and saliva differs significantly from the procedure used to analyze cultured samples. By treating the PCR positive samples as SARS-CoV-2 and the PCR negative samples as noise, there is no longer a need for the classifier to recognize the virus particles themselves – it is possible other aspects of the training set are being recognized. Comparing the PCR positive and negative current traces from nasal swab samples in Fig. 3(a), one can see clear differences. However, the differentiating characteristics seem very different to those apparent in the saliva samples in Fig. 3(b).
4. In Figure 4, only the waveforms that led to an accurate negative or positive determination are considered. Why? The basis for claiming 92% specificity and 95% sensitivity is unclear to me given the results presented.
5. The classification of nanoparticles can be done purely on the basis of particle size, and the authors explicitly state on line 45 that the viruses they consider have similar sizes. Therefore, it is unclear how the results of the nanoparticle measurements relate to the identification of viruses.

A few other issues are worth considering:

1. It's puzzling to me why the accuracy would improve when more viruses are considered.
2. The unreferenced claim that RT-PCR is prone to false negatives has not been removed, despite the authors' claim to have done so.
3. Similarly, the misleading description of the nanopore current as very feeble remains.

Overall, this manuscript has improved, but there remain sufficient concerns with the key claims that I cannot recommend its publication in Nature Communications.

Reviewer #2:

Remarks to the Author:

The author provides the missing information in this corrected version. I only suggest to the authors to add two references related to the use of the machine learning using solid-state nanopore i.e. ACS Sens. 2020, 5, 11, 3398–3403, Biosensors 2020, 10,140

Reviewer #4:

Remarks to the Author:

This revised manuscript contains a large amount of additional data that was missing in the original submitted manuscript. This additional data is very helpful in providing additional understanding as to how the proposed virus diagnostic is functioning. This additional understanding reduces some confusion but does not remove many of the main concerns that this reviewer had with the first manuscript iteration.

1) The increased data shown for beads of different sizes is very clear, and shows that for non biological particles this approach has laudable merit.

2) increased data density for virus particles diminishes this reviewer's confidence that this is a highly discriminative approach. This is highlighted with the Ip and Vd graphs, which emphasize the similarity rather than the differences of the traces.

3) additional explanation of the patient sample testing indicates that the authors do not have an independent test set outside of the training set, making it difficult to understand how their accuracy may be impacted by overfitting.

4) Minor point; Authors have not removed all of the misleading sentences in their introduction regarding the strength/weakness of rtPCR as they state they have in their rebuttal letter.

based on the data presented I remain optimistic that the authors may actually be measuring something that is meaningful. I do not have confidence that the authors understand what it is that they are measuring. There may be other nanoparticles or other sample aspects that are important but are being conflated with virus in this study.

.....

An artificial intelligence nanopore platform
for SARS-CoV-2 virus detection

.....

By Masateru Taniguchi, Shohei Minami, Chikako Ono, Rina Hamajima, Ayumi Morimura, Shigeto Hamaguchi, Yukihiro Akeda, Yuta Kanai, Takeshi Kobayashi, Wataru Kamitani, Yutaka Terada, Koichiro Suzuki, Nobuaki Hatori, Yoshiaki Yamagishi, Nobuei Washizu, Hiroyasu Takei, Osamu Sakamoto, Norihiko Naono, Kenji Tatematsu, Takashi Washio, Yoshiharu Matsuura & Kazunori Tomono

We would like to express our sincere gratitude to the referees for their valuable and helpful comments. Additional experiments were conducted according to the reviewers' suggestions. We have carefully studied all of the reviewers' concerns and incorporated the following revisions.

List of additional content included in the revised manuscript:

The following attributes are included in the main text:

1. Figure 3f. Time dependence of sensitivity and specificity in the learning process
2. Figure 3g. Time dependence of sensitivity and specificity in the diagnostic process
3. Figure 3h. Time dependence of the positive confidence in true positive specimens in the diagnostic process
4. Figure 3i. Time dependence of the positive confidence in true negative specimens in the diagnostic process
5. Figure 3j. Time dependence of the positive confidence in false positive specimens in the diagnostic process
6. Figure 3k. Time dependence of the positive confidence in false negative specimens in the diagnostic process
7. Two references

The following additional information is included in the supporting information:

1. Figure S6 and Table S4. Identification accuracy of 200 and 220 nm nanoparticles using nanopores
2. Figure S12. Time dependence of the positive confidence in true positive specimens and true negative specimens in the learning process
3. Tables S5 and S6, Figure S11. Errata for all specimens in the learning process
4. Tables S7 and S8, Figure S13. Errata for all specimens in the diagnostic process

Reviewers' comments:

Reviewer #1 (Remarks to the Author):

The inclusion of primary data has improved the revised manuscript by Taniguchi et al. Unfortunately, the raw data are still not publicly available, and this makes it impossible to ascertain the reproducibility of the results.

Response

Following the reviewer's comment, the data has been publicly available under the following link:

WEB; <https://doi.org/10.5281/zenodo.4529371>

Comment

The authors added virus the trans compartment (line 140), and later found virus in a sample drawn from the trans compartment by RT-PCR (line 143). They took this as evidence that the measured current traces were produced by virus translocations. That conclusion is unfounded.

Response

As the reviewer indicated, not all current traces can serve as evidence of the origin of the virus. Nevertheless, the RT-PCR results demonstrated that the virus passed through the nanopores. To not mislead the reader, the explanation has been revised as follows:

<Main text>

To confirm that the virus passed through the nanopore at -0.1 V, a RT-PCR measurement is performed on the solution in the *trans* channel.

To confirm that the novel coronavirus passed through the nanopore at 0.1 V, a RT-PCR measurement is performed on the solution in the *trans* channel after obtaining approximately 3,000 waveforms for both voltages.

Comment

The authors trained between 4 and 9 nanopores with waveforms from samples of the four types of virus. Clearly, different nanopores were used to train the classifier on

different viruses. The accuracy of that classifier at identifying different virus types with the same nanopore was not reported. It is therefore possible that the classifier is in fact recognizing particular nanopores, not viruses.

Response

We agree that the reviewers' concerns should be considered when conducting machine learning. Nonetheless, the BSL3 facilities that can handle SARS-CoV and SARS-CoV-2, BSL3 facilities that can handle MERS-CoV, and BSL2 facilities that can handle HCoV-229E are different. From the viewpoint of preventing contamination by other viruses, it is not possible to conduct experiments that handle these viruses simultaneously.

It is imperative to produce nanopores with high accuracy and reproducibility because machine learning may be looking at differences in the nanopores. In this work, to investigate the disparities between the nanopores as well as to determine the accuracy of discrimination between the nanoparticles at 200 and 220 nm, we used different nanopores. The results, including *F*-Value accuracy of 97% to 100% for all nanopore combinations, are shown in Figure S5. This indicated that the differences between the nanopores were small.

According to the reviewer's suggestions, we measured and identified nanoparticles at 200 and 220 nm using the same nanopore module. The corresponding results are presented in Figure S6 of the Supplementary Information. As shown in Figure S6, the average *F*-value indicating the identification accuracy was 0.98. The experimental outcomes concerning these nanoparticles imply that they identify the nanoparticles, and not the differences in the nanopore modules.

Comment

The training procedure used to classify samples from nasal swabs and saliva differs significantly from the procedure used to analyze cultured samples. By treating the PCR positive samples as SARS-CoV-2 and the PCR negative samples as noise, there is no longer a need for the classifier to recognize the virus particles themselves – it is possible other aspects of the training set are being recognized. Comparing the PCR positive and negative current traces from nasal swab samples in Fig. 3(a), one can see clear differences. However, the differentiating characteristics seem very different to those apparent in the saliva samples in Fig. 3(b).

Response

Thank you for your valuable comment. We initially attempted to diagnose clinical

specimens using cultured viruses as the learning data. However, we found that the cultured virus obtained from the utilized cell line exhibited a positive surface charge. In contrast, the new coronavirus in all clinical specimens was negatively charged. Hence, the method of using cultured viruses for learning could not be employed.

As the reviewer points out, the nasal swabs and saliva samples appear different. We are intrigued about the reasons for this variability. At present, we are attempting to measure more samples to investigate the cause of the differences between the two types of samples. However, the collection of new samples is challenging as nasal swabs are no longer performed at hospitals and health centers. Moreover, following the suggestions made by Reviewer 4, we conducted additional experiments on 180 saliva samples and performed machine learning on the learning and test data sets. Nonetheless, as mentioned above, nasal swabs are no longer collected; therefore, the data for nasal swab samples have been removed from this article.

Comment

In Figure 4, only the waveforms that led to an accurate negative or positive determination are considered. Why? The basis for claiming 92% specificity and 95% sensitivity is unclear to me given the results presented.

Response

In Figure 4 (which corresponds to Figure 3 in the revised manuscript), the positive and negative judgments were made using both correct and incorrect waveforms. The reviewer's concerns appear to arise from an algorithm in which this measurement uses a single molecule measurement to determine whether a sample is negative or positive. The judgment involves two steps. First, the accuracy (F -value) of the positive and negative judgment of each waveform obtained by a measurement is calculated. Subsequently, the positive and negative confidence of one waveform is calculated based on the judgment accuracy of all individual waveforms acquired by measuring one sample for N minutes ($N = 1-5$). The positive and negative ratios are then determined based on the confidence of all the waveforms obtained in N minutes.

The specimen sensitivity and specificity are given by the three equations shown in Figure S4.

For instance, a 5 min measurement of a saliva sample gave a confusion matrix illustrated in Figures S11 and S13.

Using the confusion matrix obtained from the 5 min measurement (Figure S13) as well as the definitions of sensitivity and specificity, the sensitivity and specificity were established to be 90% and 96%, respectively.

$$\text{Sensitivity} = 45 / (45 + 5) = 0.90$$

$$\text{Specificity} = 48 / (48 + 2) = 0.96$$

To clarify the above, Figures S11 and S13 are included in the main text of the manuscript.

Comment

It's puzzling to me why the accuracy would improve when more viruses are considered.

Response

In this study, the positive and negative judgments of samples were performed in two steps according to the algorithm shown in Figure 4 (corresponding to Figure 3 in the revised manuscript). Machine learning of the individual waveforms obtained by measurement was used to determine the judgment accuracy (F -value) of the individual waveforms. Subsequently, the waveforms that were judged to be positive or negative were assembled. The positive and negative samples were then judged based on the ratio of the positive and negative waveforms. When a sample was judged to be positive or negative, it was classified as binary. In the simplest model, where the probability of a positive is indicated by p , the accuracy of the positive and negative judgment follows the binomial theorem binomial distribution:

$$\sum_{\frac{n}{2}}^n nC_m p^m (1-p)^{n-m}$$

where p , n , and m refer to the macro F -value, number of detected pulses, and correct answers, respectively (ex. ACS Sens. 2020, 5, 11, 3398–3403).

Comment

The unreferenced claim that RT-PCR is prone to false negatives has not been removed, despite the authors' claim to have done so.

Response

We apologize for this oversight and for not removing the description of RT-PCR from the main text. We confirm that we have now deleted the claim relating to RT-PCR from the revised manuscript.

Comment

Similarly, the misleading description of the nanopore current as very feeble remains.

Response

Thank you for pointing this out. This is entirely the authors' error. The description "very feeble" has been deleted.

Reviewer #2 (Remarks to the Author):

The author provides the missing information in this corrected version. I only suggest to the authors to add two references related to the use of the machine learning using solid-state nanopore i.e. ACS Sens. 2020, 5, 11, 3398–3403, Biosensors 2020, 10,140

Response

According to the reviewer's comment, we have added the above two references.

Reviewer #4:

This revised manuscript contains a large amount of additional data that was missing in the original submitted manuscript. This additional data is very helpful in providing additional understanding as to how the proposed virus diagnostic is functioning. This additional understanding reduces some confusion but does not remove many of the main concerns that this reviewer had with the first manuscript iteration.

Comment

The increased data shown for beads of different sizes is very clear, and shows that for non biological particles this approach has laudable merit.

Response

Thank you for your comment.

Comment

Increased data density for virus particles diminishes this reviewer's confidence that this is a highly discriminative approach. This is highlighted with the I_p and V_d graphs, which emphasize the similarity rather than the differences of the traces.

Response

Overwriting of 100 waveform data of virus particles and histograms of I_p and t_d is standard in the analysis of nanopores. However, as the reviewers indicate, it is difficult to identify the four viruses with high accuracy using I_p and t_d . Hence, in this work, we analyzed the waveform data using machine learning.

Comment

Additional explanation of the patient sample testing indicates that the authors do not have an independent test set outside of the training set, making it difficult to understand how their accuracy may be impacted by overfitting.

Response

According to the reviewers' suggestions, we conducted additional experiments on 180 samples for machine learning using the training and test datasets. The training data was independent of the test data. For the training data, saliva PCR-positive samples ($n = 40$) and saliva PCR-negative samples ($n = 40$) were used. In contrast, saliva PCR-positive samples ($n = 50$) and saliva PCR-negative samples ($n = 50$) were utilized for diagnosis as

test data outside the training dataset.

In the training data, sensitivity of 100% and specificity of 100% were obtained regardless of the measurement time. When the test data was learned using the classifier obtained based on the training data, sensitivity of 90% and specificity of 96% were established following a 5 min measurement.

The saliva results obtained so far were replaced with the above outcomes. Based on this substitution, the time dependence of confidence, confusion matrix at each measurement time, and errata for saliva samples have been revised.

Notably, nasal swabs are no longer collected at hospitals and insurance offices; therefore, sample collection is challenging. Consequently, we have removed the results of the nasal swabs from this paper due to insufficient samples to create the training and test datasets.

Comment

Minor point; Authors have not removed all of the misleading sentences in their introduction regarding the strength/weakness of rtPCR as they state they have in their rebuttal letter.

Response

We apologize for this oversight and for not removing the description of RT-PCR from the main text. We confirm that we have now deleted the misleading sentences concerning RT-PCR from the revised manuscript.

Reviewers' Comments:

Reviewer #1:

Remarks to the Author:

The manuscript has been significantly improved by the latest revisions.

This reviewer appreciates the publication of raw data files; the authors are nonetheless requested to complete the online documentation so it is clear to outsiders under what conditions the various data files were obtained and in which figures the results were used.

This reviewer also appreciates the practical limitations that facility safety protocols imposed on the work. It would be worth describing those limitations and how they prevented measurements of different strains of virus from being performed with the same nanopore. That fact may have meaningfully affected the results, but it is an understandable practical limitation and one that should not prevent the publication of the results.

Two issues remain that the authors should address before the manuscript is accepted for publication:

1. Regarding the translocation of virions under positive applied voltages: It seems improper to conclude it passed through under positive voltages when RT-PCR was performed after exposing system to both positive and negative voltages. The waveforms observed under positive applied voltages could originate in the translocation of a different type of object with the opposite charge.

2. The procedures followed for computing sensitivity, specificity, and accuracy need to be explained more clearly. This reviewer was not able to follow the description provided to reproduce the results. Furthermore, the authors are taking the the F-value to be synonymous with accuracy, but this is not standard in the field. The authors should explicitly define the meaning of the measures they cite and illustrate how these were computed in the complex samples.

If those issues can be resolved, I would recommend the manuscript for publication.

Reviewer #4:

None

.....

**An artificial intelligence nanopore platform
for SARS-CoV-2 virus detection**

.....

By Masateru Taniguchi, Shohei Minami, Chikako Ono, Rina Hamajima, Ayumi Morimura, Shigeto Hamaguchi, Yukihiro Akeda, Yuta Kanai, Takeshi Kobayashi, Wataru Kamitani, Yutaka Terada, Koichiro Suzuki, Nobuaki Hatori, Yoshiaki Yamagishi, Nobuei Washizu, Hiroyasu Takei, Osamu Sakamoto, Norihiko Naono, Kenji Tatematsu, Takashi Washio, Yoshiharu Matsuura & Kazunori Tomono

We would like to express our heartfelt appreciation to the referee for his insightful remarks. We have carefully considered all of the reviewer's concerns and made the following changes.

The supporting information includes the following supplementary information:

1. Figure S1 shows a flowchart of machine learning algorithms used on nanoparticles and cultured coronaviruses
2. Figure S10. Schematic flowchart of clinical specimens
3. Tables S9. Correspondence between data in the database and samples and figures in the main text and Supplementary Information.

Reviewers' comments:

Reviewer #1 (Remarks to the Author):

The manuscript has been significantly improved by the latest revisions.

This reviewer appreciates the publication of raw data files; the authors are nonetheless requested to complete the online documentation so it is clear to outsiders under what conditions the various data files were obtained and in which figures the results were used.

We registered the correspondence table in a database in response to the reviewer's comments, indicating the correspondence between the data in the database and the sample and figures in the online documentation. In addition, we have added the online documentation from the database to Table S9 of Supplementary Information to make it easier for readers to access the data.

This reviewer also appreciates the practical limitations that facility safety protocols imposed on the work. It would be worth describing those limitations and how they prevented measurements of different strains of virus from being performed with the same nanopore. That fact may have meaningfully affected the results, but it is an understandable practical limitation and one that should not prevent the publication of the results.

Thank you for your valuable feedback. We have added the experimental limitations to the nanoparticle measurement results (Figure S7) with the same nanopores in the Supplementary Information in response to the reviewer's comment.

Two issues remain that the authors should address before the manuscript is accepted for publication:

1. Regarding the translocation of virions under positive applied voltages: It seems improper to conclude it passed through under positive voltages when RT-PCR was performed after exposing system to both positive and negative voltages. The waveforms observed under positive applied voltages could originate in the translocation of a different type of object with the opposite charge.

Thank you for your suggestions. Two types of independent experiments were conducted, one with positive voltage and the other with negative voltage. We could not get enough waveforms at a voltage of $-0.1V$. In order not to mislead the reader, the text has been revised as follows.

(Main text)

To confirm that the novel coronavirus passes through the nanopore at $0.1 V$, RT-PCR is performed on the solution in the *trans* channel after obtaining about 3,000 waveforms.

2. The procedures followed for computing sensitivity, specificity, and accuracy need to be explained more clearly. This reviewer was not able to follow the description provided to reproduce the results. Furthermore, the authors are taking the F-value to be synonymous with accuracy, but this is not standard in the field. The authors should explicitly define the meaning of the measures they cite and illustrate how these were computed in the complex samples.

We have added the definitions of sensitivity, specificity, and accuracy, as well as an explanation of the calculation procedure to the main text and Supplementary Information in response to the reviewer's comments (Figure S5). In addition, the meaning of the F -value and its calculation procedure has been added to the main text.

(Main text)

Precision in machine learning refers to the percentage of data that are expected to be positive and are actually positive, whereas recall refers to the percentage of data that is actually positive and is predicted to be positive. Since precision and recall are in a tradeoff relationship, an index that considers these two indexes together, i.e., the harmonic mean of the two indexes, is defined as the F -value. The F -value is defined by equation (4) in Figure S5 of SI and is calculated using the confusion matrix.

We have added a schematic flowchart of machine learning to Figure S1 and Figure S10 of the Supplementary Information in response to the reviewer's comment.

Figure S1. Schematic flowchart implemented on 2-types of nanoparticles and cultured coronaviruses. After acquiring the ionic current-time waveform data of the viruses (nanoparticles) of A and B, the waveform data of A and B are machine-learned to distinguish between A and B. Machine learning is performed until the highest identification accuracy is obtained using the *F*-value as an index. The *F*-value is calculated using equations (1), (2), and (4) in Figure S5.

Figure S10. Schematic flowchart of machine learning of clinical specimens. After the PCR-positive and the PCR-negative samples are measured and ionic current-time waveforms are obtained (the waveforms are machine-learned), machine learning is used until the maximum accuracy for distinguishing between the positive and negative with a single waveform is obtained using the F -value as an index. Using equations (1), (2), and (4) in Figure S5, the F -value of one waveform is calculated. Since one sample gives a large number of waveforms, a large number of waveforms obtained from one sample are used to determine whether the sample is positive or negative. Machine learning is used until the highest probability that distinguishes between the positive and negative of one sample is obtained. Sensitivity and specificity are calculated using equations (2) and (3) in Figure S5.